# Characterization and Valorization of the Agricultural Waste Obtained from *Lavandula* Steam Distillation for Its Reuse in the Food and Pharmaceutical Fields

**DOI:** 10.3390/molecules27051613

**Published:** 2022-02-28

**Authors:** Eleonora Truzzi, Mohamed Aymen Chaouch, Gaia Rossi, Lorenzo Tagliazucchi, Davide Bertelli, Stefania Benvenuti

**Affiliations:** 1Department of Life Sciences, University of Modena and Reggio Emilia, Via G. Campi 103, 41125 Modena, Italy; eleonora.truzzi@unimore.it (E.T.); mohamedaymen.chaouch@unimore.it (M.A.C.); 214201@studenti.unimore.it (G.R.); lorenzo.tagliazucchi@unimore.it (L.T.); davide.bertelli@unimore.it (D.B.); 2Doctorate School in Food and Agricultural Science, Technology and Biotechnology (STEBA), University of Modena and Reggio Emilia, 42122 Reggio Emilia, Italy; 3Doctorate School in Clinical and Experimental Medicine (CEM), University of Modena and Reggio Emilia, 41125 Modena, Italy

**Keywords:** lavender, sustainable agriculture, waste recovery, circular economy, antioxidants

## Abstract

The main focus of the current research was the characterization of the by-products from the steam distillation of *Lavandula angustifolia* Mill. (LA) and *Lavandula x intermedia* Emeric ex Loisel (LI) aerial parts, as they are important sources of bioactive compounds suitable for several applications in the food, cosmetic, and pharmaceutical industries. The oil-exhausted biomasses were extracted and the total polyphenol and flavonoid contents were, respectively, 19.22 ± 4.16 and 1.56 ± 0.21 mg/g for LA extract and 17.06 ± 3.31 and 1.41 ± 0.10 mg/g for LI extract. The qualitative analysis by liquid chromatography-electrospray tandem mass spectrometry (HPLC-ESI-MS) revealed that both the extracts were rich in phenolic acids and glycosylated flavonoids. The extracts exhibited radical scavenging, chelating, reducing activities, and inhibitory capacities on acetylcholinesterase and tyrosinase. The IC50 values against acetylcholinesterase and tyrosinase were, respectively, 5.35 ± 0.47 and 5.26 ± 0.02 mg/mL for LA, and 6.67 ± 0.12 and 6.56 ± 0.16 mg/mL for LI extracts. In conclusion, the oil-exhausted biomasses demonstrated to represent important sources of bioactive compounds, suitable for several applications in the food, cosmetic, and pharmaceutical industries.

## 1. Introduction

In March 2020, the European Commission adopted the “new circular economy action plan”, one of the main building blocks of the “European green deal” for sustainable growth [1]. The action plan promotes the circular economy and ensures the prevention of waste production, supporting the regeneration of resources. The program was born to face the increment of agricultural wastes, co-products, and by-products in Europe, which have been esteemed to be more than 700 million tons every year [2]. In this context, the scientific community is utilizing efforts to valorize the food and agricultural wastes, which might be rich sources of valuable compounds. Nowadays, these wastes, also called biomasses, are mainly used for energy recovery by producing biofuel. However, this is the less preferable procedure to manage the biomass, according to the “waste hierarchy” proposed by the Environmental Protection Agency (EPA) [3]. Based on the EPA’s scheme, waste reuse and recycling are favored over recovery energy. Plant and food biomasses are rich in polysaccharides (such as pectin and cellulose), enzymes (such as bromelain from pineapples), and secondary metabolites. Secondary metabolites are essential for plant long-term survival and can act as antimicrobial agents, protectors against predators, or attractors of pollinators [4]. Since ancient times, the positive effects of secondary metabolites on human health were observed through the consumption of herbs in traditional medicine. These compounds are usually classified according to the biosynthetic pathway, and the polyphenols’ group is one of the main ones, along with alkaloids and terpenes. The polyphenols group includes phenols, phenolic acids, flavonoids, and tannins, which are in turn classified based on the hydroxyl groups, the number of aromatic rings, and the number of carbon atoms [5]. Among the several beneficial activities of such compounds, the antioxidant, anti-inflammatory, and antimicrobial are the most recognized and studied [6,7,8,9,10]. For these properties, polyphenols exhibit health benefits for treating and preventing several conditions, such as age-related diseases, heart diseases, and cancers.

Essential oils (EOs) are obtained by steam distillation of flowers, roots, leaves, and fruit peels of aromatic plants. Due to the countless activity and properties, the EOs are commonly used in several industrial fields. Recent reports highlighted that the global market of the EOs is growing due to consumers’ higher attention to “green” products, where synthetic active compounds are more and more replaced by natural compounds in cosmetics and foods [11]. In addition, the global market is destined to grow since the novel potentiality of the EOs is arising. Nowadays, the oil-exhausted biomass obtained after the extraction of EOs is considered waste, mainly used for mulching in other crop cultivations or for energy production from combustion. However, the biomass is still rich in polyphenols which are not volatile and thus not collected in the EO. For these reasons, the valorization of the biomasses from the production of EOs is an important issue to promote recycling and to increase the value of these crops. Furthermore, being the majority of the EOs approved for food flavoring, the waste and by-products of their production can be employed in organic farming as a feed additive, according to the European Council Regulation 2018/848 [12].

The *Lavandula* genus is cultivated worldwide for its EOs, which are largely employed in cosmetics, food processing, perfumes, aromatherapy, and drugs. According to Giray’s report, the production of lavender EOs is dynamically increasing, following today′s trend in “natural and organic” [13].

*Lavandula* genus includes several species, and the most used in the industries are the *L. angustifolia* (LA) and *L. × intermedia* (LI). Lavender can be considered one of the most produced EO, especially in Bulgaria, France, Italy, China, and Spain [13]. Furthermore, lavender EO is considered one of the most valuable EOs, despite numerous cases of adulteration which led the regulatory organization to promote interventions to guarantee their authenticity, including through the development and application of new analytical methods in the last few years [14,15,16,17,18,19,20]. Indeed, lavender EOs demonstrated to be extremely versatile due to their different biological activities. In the last decades, lavender EOs have been demonstrated to act on the central nervous system, exerting anti-depressive, anxiolytic, antioxidant, and anti-inflammatory effects both in humans and animals [21,22,23,24,25,26]. Furthermore, regarding the agro-food industry, these EOs showed promising antibacterial activity on antibiotic-resistant bacteria and fungi [27,28] and exhibited pronounced effects on different pests and weeds [29,30,31,32]. For these reasons, lavender EOs might be also employed in food safety and stability.

Thus, the present work was focused on the characterization of the solid waste from the steam distillation of LA and LI, as a source of bioactive compounds. Several biological activities were evaluated to explore the potentiality of the reuse of lavender biomass. To the best of our knowledge, this is the first study that aimed at the evaluation of the enzyme inhibitory capability of extracts obtained from oil-exhausted lavender biomasses.

## 2. Results and Discussion

Lavender EOs are among the most produced EOs in Europe, and the exhausted biomass obtained from the steam distillation is currently considered a low valuable by-product. To promote the reuse of this agricultural waste, an extensive study of the chemical composition and the activities of the bioactive compounds still present after the steam distillation is required. Indeed, it is certainly worth remarking that the biomass wastes from the steam distillation process underwent thermal stress due to the high temperatures employed for the extraction of the EOs. As a consequence, the bioactive compounds present in plant biomasses might be degraded and chemically modified, resulting in a decrease in their biological activities. For this reason, the study of the biological and chemical activities of polyphenols extracted from lavender must be performed on oil-exhausted biomass to demonstrate the importance and potentiality of the reuse of this waste. Thus, in the present work, the exhausted biomasses of *L. angustifolia* and *L. intermedia* were considered as the main characters of the research, as sources of natural bioactive compounds. In that, the extracts of LA and LI were characterized in terms of composition and antioxidant and inhibitory capacities against certain enzymes. The methods and results concerning the EOs and the hydrosols obtained from the steam distillation process are fully described in the Appendix A. Specifically, the chemical composition of the Eos and hydrosols is reported in Appendix A, respectively.

### 2.1. Total Phenolic and Total Flavonoid Content

The solvent screening for the recovery of polyphenols from oil-exhausted lavender biomasses suggested that the mixture of ethanol and water (50:50) was the optimal solvent in terms of TPC. The yields of the dry extracts were determined by freeze-drying the liquid extracts, obtaining 211.5 ± 7.8 mg/g e 193.0 ± 2.6 mg/g of dry extract of biomass for LA and LI, respectively.

The estimation of the total phenolic content (TPC) and total flavonoid content (TFC) was carried out through Folin–Ciocalteu and aluminum chlorate assays, respectively. The two hydroalcoholic extracts did not show any significant difference in both TPC and TFC contents (Table 1).

The TFC represented about 8% of the TPC in both lavender extracts. In the literature, different results on the *Lavandula* genus can be observed, and the diversity might be due to several factors.

As an example, Spiridon et al. reported higher values for the alcoholic extract from *Lavandula angustifolia* leaves and flowers. The TPC and TFC were found to be 50.6 ± 3.16 mg GAE/g and 27.6 ± 3.42 mg Rutin equivalent/g, respectively [33]. On the other hand, Duda et al. (2015) showed similar TPC and TFC results by studying the whole biomass of *L. angustifolia* harvested in two different phenological periods (the beginning of flowering and the full bloom). They observed a TPC of between 12.44 and 18.16 mg GAE/g, and a TFC between 3.37 and 4.85 mg QE/g dry plant [34]. These differences might be due to the part of the biomass considered in the study. Indeed, by comparing the results from these studies, the leaves and the flowers of lavender seem to be richer in polyphenols than stems, which are evaluated in the extraction of the entire biomass. In addition, it has been reported that the TPC of lavender is strongly affected by the species, harvest time, growing conditions of the crops, and plant age.

By comparing the results obtained in this study with other works concerning lavender wastes from the distillation process, Turrini et al. observed higher levels of both TPC and TFC (40.15 ± 0.04 mg GAE/g and 4.72 ± 3.56 mg QE/g, respectively) in *Lavandula angustifolia* biomass after pulsed ultrasound-assisted extraction [35]. On the contrary, Slavov et al. noticed lower TPC that ranged from 7.52 and 10.75 mg GAE/g in Bulgarian lavender (*Lavandula angustifolia*) waste [36]. Moreover, the obtained values were higher than those obtained in the research conducted by Méndez-Tovar et al. on the *Lavandula latifolia* EO distillation by-product. The TPC content in the studied samples varied between 1.89 ± 0.09 and 3.54 ± 0.22 mg GAE/g of dry flowers [37].

The dry extracts contained about 9% and 0.7% of TPC and TFC, respectively. These results suggested that the majority of the extracts were composed of other substances, such as fibers, lignin, organic acids, triterpenoids, and sugars [38,39].

### 2.2. LC−ESI−MS and MS/MS Analysis

To understand the composition of lavender extracts, an LC-ESI-MS analysis was carried out. The detected compounds in the hydroalcoholic extract of the residual plant material were identified using data acquired by LC-ESI-MS of the parent ions and data-dependent MS/MS fragmentation. A typical chromatogram of lavender extracts is displayed in Figure 1. The retention times, molecular ions, fragmentation patterns, tentative identifications, molecular weights, and formulas are illustrated in Table 2. LC-ESI-MS and MS/MS analyses allowed the detection of the molecular ion for each compound and produced the fragmentation in negative mode. Chlorogenic acid, caffeic acid, 4-coumaric acid, ferulic acid, rosmarinic acid, luteolin, and apigenin standard compounds were used for the identification of the aglycones. The tentative identification of glycosides was supposed basing only on the fragmentation of precursor ions.

In general, the LA extract exhibited a higher abundance of phenolic acids and flavonoids. The main compounds found in the extracts were also reported by other authors [35,38,39].

Among the phenolic acids, the most abundant were the derivatives of caffeic acid, p-coumaric acid, ferulic acid, and rosmarinic acid. Compound 1 was identified as caffeoyl aspartic acid due to the fragment at *m*/*z* 179 with a relative loss of 115 Da, which corresponds to the aspartic acid moiety [40]. Compound 2 was recognized as danshensu (3,4-dihydroxyphenyl lactic acid), the derivative of caffeic acid, which showed the precursor ion of *m*/*z* 197 and the fragments *m*/*z* 178.9 and 135, caused by the loss of hydroxylic (18 Da) and carboxylic (44 Da) groups, respectively [41]. Compounds 4, 8, and 6, 11 were classified as hexose-derivative of *p*-coumaric acid and ferulic acid, respectively, due to the loss of a glucosyl moiety (162 Da) and the characteristic fragment at *m*/*z* 119 and *m*/*z* 149 (loss of a carboxylic group, 44 Da) [42]. Finally, the unprotonated molecular ion at *m*/*z* 359.1 was identified as rosmarinic acid (compound 19). The fragments at *m*/*z* 196.9 and 178.9 were related to the loss of danshensu and caffeic acid moieties, while the fragment at *m*/*z* 160.9 to the following loss of one water molecule [43]. Compound 22 was tentatively identified as a derivative of rosmarinic acid due to the presence of the fragments *m*/*z* 179 and 135 [44,45].

Regarding the flavonoids, the identified compounds were derivatives of quercetin, apigenin, luteolin/kaempferol. Luteolin and kaempferol could not be distinguishable due to the same fragmentation patterns. The derivatives were identified due to the loss of glucosidic (162 Da), rhamnosidic (146 Da), and glucuronic (176 Da) moieties. The aglycones were classified based on the characteristic product ions. Luteolin and kaempferol derivatives exhibited the fragment at *m*/*z* 284.9 [42], quercetin—fragments at *m*/*z* 301, 179, and 151 [46], and apigenin—fragments *m*/*z* 289 and 175 [47]. Finally, compound 29 was tentatively identified as ellagic acid, due to the precursor ion at *m*/*z* 301 and the product ion at *m*/*z* 283 [48].

### 2.3. Antioxidant Activity

The antioxidant properties of the two extracts were evaluated by calculating the direct neutralization of free radicals generated by DPPH, and the prooxidant activities related to the interaction with iron ions. The prooxidant activity is exerted by the chelation of iron (II) and the reduction of iron (III), both involved as a catalyst in the Fenton reaction. Indeed, the chelation or reduction of iron ions prevents the conversion of hydrogen peroxide to hydroxyl radicals [49]. The IC50 values of the DPPH inhibition were 0.17 ± 0.02 and 0.17 ± 0.01 mg of biomass for LA and LI, respectively. The amount of the standard reference Trolox that gave the same inhibition was 16.40 ± 0.41 μg. Regarding the iron chelation, the IC50 values were 22.17 ± 0.42 mg of biomass, 15.77 ± 0.10 mg, and 33.00 ± 0.34 μg of biomass from LA and LI, and EDTA, respectively. The reducing power of the extracts was calculated as described in the Methods section and was esteemed equal to that obtained by the AA solutions at a concentration of 0.313 ± 0.014 and 0.261 ± 0.010 mg/mL for LA and LI, respectively.

To evaluate the antioxidant strength of lavender biomasses, the mg equivalents of the references per gram of biomass were calculated (Table 3).

The results highlighted the marked antioxidant properties of the extracts of lavender biomasses. This evidence was in contrast to Miliauskas et al.’s findings, where acetone extract of LA did not exhibit remarkable antioxidant activities [50]. Conversely, several other authors highlighted strong dose-dependent scavenging, chelating, and reducing activities of lavender and lavandin extracts [35,36,51,52]. Specifically, the two lavenders exhibited similar free scavenging activities, while the chelating and reducing activities demonstrated opposite trends. LA showed a significantly higher content of eqAA/g (*p* < 0.0001), suggesting a greater capability in reducing ferric ions, as reported by Blažeković et al. [53]. The higher activity of LA compared to LI in oxidation–reduction reactions resulted in an agreement with the greater total phenolic content (Table 1). In addition, besides the phenolic acids and flavonoids, several other compounds might contribute to the antioxidant power of the extracts, such as organic acids which have been reported in the *Lavandula* genus [39]. On the contrary, LI exhibited a significantly higher activity (*p* < 0.01) in chelating ferrous ions, correlated to the major content of eqEDTA/g. Similar evidence was also reported by Robu et al., where LI biomass displayed a greater chelating activity than LA biomass [54]. This result might be due to the major concentration of polyphenols with more than one chelating site or with greater stability constants of the complex. Indeed, the metal chelation potential of polyphenols is strongly related to the catechol moieties and the combination of hydroxyl and carbonyl groups, characteristic of the flavonoid structure [55]. Therefore, even though the TPC and TFC of LI were slightly smaller than LA, these results suggested the presence of a higher concentration of stronger chelating polyphenols, such as rosmarinic acid, luteolin, and kaempferol.

### 2.4. In Vitro Acetylcholinesterase (AChE) and Tyrosinase Inhibition Assay

Nowadays, with cholinesterase inhibition being the most widely used approach for the treatment of Alzheimer’s disease, several efforts have been made to discover new sources of inhibitors. Different plants have been tested to understand their effects on AChE [56,57,58]. Indeed, flavonoids and phenolic acids have been reported to fit into the gorge of the active site of the enzyme [59]. Furthermore, these compounds demonstrated strong tyrosinase inhibiting properties, conferring them with features for several applications in the food, cosmetic, and pharmaceutical industries [60,61,62]. Indeed, tyrosinase is a widespread enzyme in food, fungi, bacteria, and animals. Tyrosinase is the enzyme responsible for food browning, and in humans, it causes melanogenesis and skin pigmentation [62].

The freeze-dried extracts of LA and LI were tested to evaluate their anti-cholinesterase and anti-tyrosinase activities (Table 4). Both extracts were demonstrated to be effective in the inhibition of the enzymes. In particular, LA extracts showed significantly lower IC50 values than LI extracts, suggesting a stronger inhibition capability (*p* < 0.01 and 0.001 for AChE and tyrosinase, respectively). The higher inhibition capacity of the LA extract might be related to the highest content of both polyphenols and flavonoids.

In the literature, the studies that aimed at evaluating the activity of *Lavandula* on AChE employed the whole fresh aerial parts of the plant to prepare the extracts. Thus, the extracts were composed of both volatile terpenes and polyphenols. No studies aimed at the evaluation of the inhibitory activity of lavender biomasses that were subjected to steam distillation prior to the extraction of polyphenols. With terpenes from EOs being well-recognized AChE inhibitors [63,64,65], a direct comparison with the results of other authors might be difficult. Vladimir-Knežević and co-authors evaluated the anti-cholinesterase capacity of ethanolic extracts of medicinal plants from *Lamiaceae* family. In their work, *Lavandula angustifolia* showed inhibition of 50% at a concentration of 1 mg/mL, while galantamine exhibited an IC50 value of 0.122 μg/mL. In addition, the authors highlighted the essential role of certain polyphenols in the inhibition in combination with the terpenes of the EOs [66]. In another report, Costa et al. affirmed that supercritical fluid extracts of *Lavandula viridis* exerted an IC50 value of 1.975 mg/mL, proving a central role of the monoterpenes of the EO. In their study, the author stated that the IC50 of the reference standard galantamine was 2.20 μg/mL under the same test condition of *L. viridis* extract [67].

Regarding the anti-tyrosinase activity of *Lavandula* extracts, no studies aimed at the evaluation of the activity of extracts obtained from by-products of the steam distillation. The only study present in the literature considered the whole fresh plant. Hsu and co-workers tested water extracts of different species of *Lavandula*, demonstrating that the strength of the inhibition was species dependent. Furthermore, in contrast with our results, all the inhibitory capacities of the extracts were impaired by the freeze-drying process. The authors explained this evidence by suggesting that the inhibitory effects of the extracts were related to the action of the enzyme polyphenol esterase, which degraded during the drying process [68].

## 3. Materials and Methods

### 3.1. Sample Materials and Chemicals

One sample of *Lavandula angustifolia* and one sample of *Lavandula x intermedia* cultivar “Grosso” aerial parts were provided from two different farms located in the Italian Tuscan-Emilian Apennines (9X4J + 7W map and 7XWH + 3F map, respectively). The aerial parts of the plants were hand-picked when the inflorescences were in full blooming during summer 2021. 1,1-diphenyl-2-picrylhydrazyl (DPPH•), quercetin, gallic acid, sodium sulphate (Na_2_SO_4_), ferrozine, iron (III) and iron (II) chlorides, ethylenediaminetetraacetic acid (EDTA), ascorbic acid, 6-hydroxy-2,5,7,8-tetramethylchroman-2-carboxylic acid (Trolox), trichloroacetic acid, potassium ferricyanide, Folin–Ciocalteu reagent, acetylcholinesterase (AChE) (electric eel, E.C. 3.1.1.7, type VI-S), acetylthiocholine iodide (ATCI), 5,5′-dithiobis (2-nitrobenzoic acid) (DTNB), galantamine hydrobromide, tyrosinase (mushroom, E.C. 1.14.18.1), L-tyrosine, kojic acid, aluminum chloride, and C_8_–C_40_ n-alkanes were purchased from Sigma-Aldrich (Milan, Italy).

Acetonitrile (ACN), acetic acid (HAc), ethylacetate (EtOAc), *n*-hexane (Hex), and ethanol were of LC–MS purity grade (Sigma-Aldrich) (Milan, Italy).

### 3.2. Lavender Steam Distillation

The EOs from lavenders were extracted from fresh aerial parts by steam distillation according to the European Pharmacopoeia X Ed., as described in our previous work [69]. Briefly, about 400 g of flowers was steam distilled for 1 h by a stainless-steel distiller coupled with a Clevenger-type apparatus (Albrigi Luigi s.r.l., Stallavena, VR, Italy). The EO collected was separated from hydrosol and measured on an analytical scale. The percent yield of the EOs was calculated as the weight of oil per weight of fresh lavender flowers. The EOs were stored at 4 °C until analysis. The oil-exhausted biomasses were collected and dried at room temperature.

### 3.3. Plant Material and Extraction Procedure

The extraction was performed by dynamic maceration, and several extracting solvents or solvent mixtures were tested (ethanol, methanol, water, ethyl acetate, and hydroalcoholic solutions at different ratios). The optimal solvent was selected based on the efficiency in recovering polyphenols, quantified by Folin–Ciocalteu method as described below. Briefly, 3.5 g of oil-exhausted aerial parts was extracted by dynamic maceration with 40 mL of the extracting solvent. The solution was filtered into a volumetric flask and the biomass was extracted two more times with 35 mL of the same solvent. The filtrates were adjusted at a final volume of 100 mL and stored at 4 °C. The extraction was performed in triplicate. For the enzymatic assays, the ethanol was removed under vacuum and the remaining aqueous suspension was freeze-dried (Lio 5P, CinquePascal, Milan, Italy). The extracts obtained by the optimal solvent (ethanol 50%) for the recovery of polyphenols were used for the following analyses.

### 3.4. Total Polyphenolic and Flavonoid Content

The total polyphenolic content (TPC) in each *Lavandula* oil-exhausted biomass was determined by Folin–Ciocalteu method. Briefly, 50 μL of the extract was mixed with 2.5 mL of 10% Folin−Ciocalteu reagent. Then, 2 mL of Na_2_CO_3_ saturated solution was added and the reaction mixture was incubated at 50 °C for 15 min. Finally, the absorbance of the solution was measured at 760 nm by using a UV/Vis spectrophotometer (UVmini-1240; Shimadzu Corp., Kyoto, Japan). The concentration of total polyphenolic compounds was calculated by using a standard curve prepared with gallic acid solutions (Appendix A). The total polyphenolic content was expressed as milligrams of gallic acid equivalents (GAE) per gram of lavender flowers. The results were expressed as the mean ± standard deviation calculated from the results obtained in duplicate for each replicate of extract (*n* = 3).

The total flavonoid content (TFC) was determined according to the aluminum chloride method in each *Lavandula* oil-exhausted biomass. Briefly, 100 µL of the sample was mixed with 1.9 mL of ethanol and 2 mL of 2% AlCl_3_ solution. The reaction mixture was incubated for 30 min at room temperature in the dark and the absorbance was measured at 420 nm by a UV/Vis spectrophotometer (UVmini-1240). The concentration of total flavonoids was determined by using a standard curve prepared with quercetin solutions (Appendix A). The total flavonoid content was expressed as milligrams of quercetin equivalents (QE) per gram of lavender aerial parts. The results were expressed as the mean ± standard deviation calculated from the results obtained in duplicate for each replicate of extract (*n* = 3).

### 3.5. Identification of Polyphenols by LC−ESI−MS and MS^2^

The liquid extracts were properly diluted and analyzed for the identification of the active compounds.

The LC-ESI-MS and MS^2^ analyses were carried out using an Agilent Technologies modular 1200 system coupled to an Agilent 6310A ion trap mass analyzer with an ESI ion source (Agilent, Waldbronn, Germany). HPLC analyses were performed on an Ascentis C18 column (250 mm × 4.6 mm I.D., 5 μm, Supelco, Bellefonte, PA, USA), with a mobile phase composed of (A) 0.3% acetic acid in water and (B) ACN. The gradient elution was set as follows: 0 min, 17% (B); 35 min, 23% (B); 52 min, 49% (B). The flow rate was set at 1 mL/min and the injection volume was 20 μL. The ESI source operated in negative ionization mode and the experimental parameters were set as follows: the capillary voltage was 3.5 kV, the nebulizer (N_2_) pressure was 32 psi, the drying gas temperature and flow were 350 °C and 10 L/min, respectively, and the skimmer voltage was 40 V.

Agilent 6300 Series Ion Trap LC/MS system software (version 6.2) was used for instrument control, data acquisition, and qualitative analysis. The mass spectrometer was operated in full-scan mode in the *m*/*z* range 200–1200. MS2 spectra were automatically performed by using the SmartFrag function with helium as the collision gas in the *m*/*z* range 50–1500.

### 3.6. Evaluation of Antioxidant Activity

The antioxidant activities were evaluated on the three different extracts obtained from each *Lavandula* oil-exhausted biomass.

#### 3.6.1. Determination of DPPH Free Radical-Scavenging and Fe^2+^ Chelating Activities

For the DPPH free radical-scavenging activity, the freshly prepared extracts were diluted (1:10) with water:ethanol (50:50) solution and different aliquots of the obtained solution (ranged from 50 μL to 1.2 mL) were further diluted with ethanol to a final volume of 2.7 mL directly in a cuvette. To each extract dilution, 300 μL of 0.04% DPPH ethanolic solution was added and the reaction mixtures were left to stand at room temperature for 15 min in the dark. The DPPH solution was freshly prepared daily and stored in a flask covered with aluminum foil in the dark at 4 °C. A DPPH control sample (containing 2.7 mL of ethanol and 300 μL of DPPH solution) was prepared and measured daily. Finally, the absorbances were measured at 517 nm against blank extracts (without the addition of DPPH) by using a UV/Vis spectrophotometer (UVmini-1240, Shimadzu Corp., Kyoto, Japan). Ethanolic solutions with different Trolox concentrations (ranged from 0.2 to 1.6 mM) were analyzed as a positive control.

For the determination of Fe^2+^ Chelating activity, to different aliquots (0.1–1.2 mL) of freshly prepared extracts, 200 μL of 2 mM FeCl_2_ solution and 200 μL of 5 mM ferrozine solution were added. The solutions were diluted with MilliQ water to 10 mL in a volumetric flask and left to stand at room temperature for 10 min. The control sample was prepared in the same manner without the addition of the extract. Finally, the absorbances were measured at 562 nm against blank extracts (without the addition of FeCl_2_ and ferrozine solutions) by using a UV/Vis spectrophotometer (UVmini-1240). EDTA was selected as positive control and different concentration (0.25–1.00 mg/mL) were analyzed. 

The DPPH scavenging and metal chelating effects were calculated as follows:(1)$$\left(inhibition\;\%\right)=\frac{\left(A_{Control}-A_{Sample}\right)}{A_{Control}}\times 100$$
where *A_Control_* is the absorbance of the control reaction and *A_sample_* is the absorbance of the sample. The free radical scavenging and the metal-chelating capacities were expressed by IC50 values extrapolated from the dose–response curves.

#### 3.6.2. Reducing Power Activity

The reducing power activity was performed according to the method by Papotti et al. with slight modifications [70]. The freshly prepared extracts were diluted (1:10) with water:ethanol (50:50) solution and different aliquots of the obtained solution (ranged from 100 to 500 μL) were further diluted with the same solvent up to 500 μL. Then, 2.5 mL of phosphate buffers solution (pH 6.6) and 2.5 mL of potassium ferricyanide 1% solution were added, and the solutions were incubated at 50 °C. After 20 min, 2.5 mL of trichloroacetic acid 10% solution, 8 mL of water, and 1.6 mL of Iron (III) chloride 0.1% solution were added. Finally, 2 mL of the solutions was diluted with 2 mL of water, and the absorbances were measured at 700 nm. The slope of the dose–response curves indicated the reducing power of the extracts. Solutions of ascorbic acid (AA) with different concentrations (100–750 μg/mL) were prepared and analyzed as described above. The slope of the dose–response curves obtained for each ascorbic acid solution was plotted against the concentration, and the equation of the linear regression curve was used to determine the reducing power of the extracts in terms of concentration of AA.

### 3.7. Acetylcholinesterase and Tyrosinase Inhibitory Assays

The freeze-dried extracts of biomass of LA and LI (three replicates for each biomass) were dissolved in PBS at the concentration of 10 mg/mL and different dilutions were prepared in the range of 0.5–10.0 mg/mL. For the inhibition of AChE, the extracts and reagents were solubilized in PBS 100 mM at pH 8. For the inhibition of tyrosinase, PBS 20 mM at pH 6.8 was employed to prepare the solutions.

The capacity of the extract in inhibiting AChE was evaluated according to Costa et al. with minor modifications [67]. The reaction solution was prepared by mixing 1 mL of DTNB 15mM, 200 μL of ATCI 3mM, 400 μL of PBS, and 200 μL of inhibitor solution (or PBS in the case of the enzymatic control) into a 1 mL cuvette. Then, 200 μL of AChE 0.115 U/mL was added, and the reaction was monitored for 5 min by recording the absorbance at 405 nm every 14 s using a UV/vis spectrophotometer (Jasco V-730, Easton, MD, USA). The absorbances were recorded against a blank solution composed of all the reactive without the enzyme. Galantamine was selected as a reference inhibitor and was tested in the range of 7–170 μg/mL under the same operative conditions.

The inhibition of tyrosinase was evaluated according to Fiocco et al. with minor changes [71]. The reaction solution was prepared by mixing 250 μL of tyrosine 1.66 mM, 700 μL of PBS (20 mM, pH 6.8), and 200 μL of inhibitor solution (or PBS in the case of the enzymatic control) in a 1 mL cuvette. Then, 300 μL of tyrosinase 170 U/mL was added, and the reaction was monitored for 40 min by recording the absorbance at 475 nm every 14 s. Kojic acid was selected as a reference inhibitor and was tested in the range 7–70 μg/mL under the same operative conditions.

For both assays, the velocities (slopes, OD/min) of the reactions were calculated for each inhibitor concentration tested, and the inhibition percentage was calculated as follows:(2)$$inhibition\;\%=\frac{\left(Slope_{CTRL\;}-Slope_{inhibitor}\right)}{Slope_{CTRL}}\times 100$$
where *Slope_CTRL_* and *Slope_inhibitor_* are the velocities of the enzyme in the absence or presence of the inhibitor, respectively. The percentages of inhibition were plotted against the concentrations of the inhibitor and the curve was fitted to calculate the IC50 value.

### 3.8. Statistical Analysis

Student’s *t*-test was used to highlight significant differences between the two lavenders (*p* < 0.05).

## 4. Conclusions

The antioxidant and enzyme inhibitory capabilities of *Lavandula* extracts have been extensively studied in the last few decades. Thus, it is well known that the aerial parts of the *Lavandula* species are important sources of polyphenols with countless activities. However, to promote the reuse and valorization of agricultural wastes from the production of lavender EOs, studies focusing on the characterization of oil-exhausted biomasses are required. Indeed, during the steam distillation process, the aerial parts of lavender are subjected to high temperatures for hours, causing the partial degradation and deactivation of polyphenols. In the present work, *L.angustifolia* and *L. intermedia* oil-exhausted biomasses were demonstrated to represent an interesting source of bioactive compounds, even though they might have been partially lost during the steam distillation process. The selected method of extraction of the oil-exhausted biomasses proved to be a promising strategy for the recovery of polyphenols by using food-grade solvents (ethanol and water), also applicable on a large scale. Several properties of lavender extracts from oil-exhausted biomass have been demonstrated, and for the first time, their enzyme inhibitory effects were evaluated. These results confer to these extracts their suitability in different fields. Indeed, the antioxidant and anti-tyrosinase activities might be exploited in the food and cosmetic industries to prevent the browning and degeneration of active compounds and ameliorate the conservation of the final products. Furthermore, these extracts might be used by the pharmaceutical industry also due to their anti-enzymatic capabilities demonstrated here. In that, they might represent a valid therapeutic alternative for the prevention and treatment of Alzheimer’s disease, hyperpigmentation, and other chronic diseases where radicals play a central role.

## Figures and Tables

**Figure 1 molecules-27-01613-f001:**
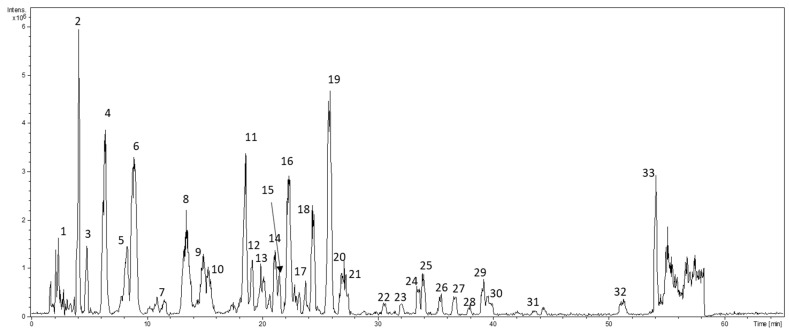
LC-MS base peak chromatogram of *Lavandula angustifolia* hydroalcoholic extract.

**Table 1 molecules-27-01613-t001:** Total phenolic content (TPC) and total flavonoid content (TFC) of *L. angustifolia* (LA) and *L. × intermedia* (LI) oil-exhausted biomasses. The results are expressed as the mean ± standard deviation of mg equivalents of gallic acid (GAE) and quercetin (QE) respectively per g of biomass (three extracts for each biomass). The TPC and TFC were calculated also as a percentage (*w*/*w*) in the dry extract.

Residual Material	LA	LI
TPC (mg GAE/g)	19.22 ± 4.16	17.06 ± 3.31
TPC % (GAE dry extract)	9.09 ± 0.33	8.84 ± 0.12
TFC (mg QE/g)	1.56 ± 0.21	1.41 ± 0.10
TFC % (QE/dry extract)	0.74 ± 0.03	0.73 ± 0.01

**Table 2 molecules-27-01613-t002:** LC-ESI-MS and MS/MS data (negative ionization mode) of the tentatively identified compounds in *L. angustifolia* (LA) and *L. intermedia* extracts. The symbols “+” and “−” indicate the presence or the absence of the compounds in the extracts.

Peak Number	Rt (min)	Tentative Identification	[M-H]^−^ (*m*/*z*)	Fragments (*m*/*z*)	Molecular Weight (g/mol)	LA	LI
1	2.3	Caffeoyl aspartic acid	294.1	179.0	295.24	+	−
2	4.2	Danshensu	395.0 (2M-H), 197.0	178.9, 135.0	198.17	+	+
3	4.9	Unknown	501.0	336.9, 295.0		+	+
4	6.5	p-coumaric acid hexose	651.3 (2M-H), 325.0	162.9, 119.0	326.10	+	+
5	8.2	Unknown	387.2	369.2, 207.0		+	+
6	8.9	Ferulic acid hexose	711.3 (2M-H), 355.1	192.9, 148.9	356.32	+	+
7	11.5	Unknown	351.0	248.9, 231.0, 177.0, 113.0		+	−
8	13.3	p-coumaric acid hexose	651.0 (2M-H), 325.1	162.9, 119.0	326.10	+	−
9	14.7	Luteolin 7-*O*-diglucuronide	637.2	461.1, 284.9	638.11	+	−
10	15.5	Apigenin 7-*O*-diglucuronide	621.0	445.1, 268.9	622.12	+	−
11	18.4	Ferulic acid hexose	711.0 (2M-H), 355.1	192.9, 149.0	356.32	+	+
12	19.2	Unknown	521.2	358.9, 229.0, 285.0		+	−
13	20.7	Quercetin hexose	463.0	301.0, 178.9	464.09	+	−
14	21.0	Luteolin/kaempferol hexose	447.2	284.9	448.10	+	+
15	21.5	Unknown	441.2	395.3, 262.9		+	+
16	22.3	Luteolin/kaempferol glucuronide	461.1	284.9	462.40	+	+
17	23.7	Quercetin 3-*O*-rhamnoside	447.1	300.9, 151.0	448.10	+	−
18	24.3	Apigenin 7-*O*-glucoside	431.2	269.0	432.40	+	+
19	25.8	Rosmarinic acid	359.1	222.8, 196.9, 178.9, 160.9,	360.31	+	+
20	26.7	Luteolin/kaempferol glucuronide	461.0	285.0	462.40	+	+
21	27.2	Apigenin 7-*O*-glucurunide	445.1	269.0, 174.9	446.40	+	+
22	30.5	Rosmarinic acid methylester	373.0	178.9, 135.0	374.30	+	−
23	31.9	Kaempferol/Luteolin	285.1	254.8, 226.9	286.05	+	−
24	33.4	Unknown	493.0	295.0, 269.1		+	+
25	33.9	Unknown	618.4	582.4, 462.3		+	−
26	35.3	Quercetin hexose	463.2	301.0	464.09	+	+
27	36.6	Unknown	507.3	345.2, 299.2		+	+
28	37.9	Unknown	329.2	221.0, 193.0, 170.9		+	+
29	39.1	Ellagic acid	301.2	283.4	302.19	+	−
30	39.6	Unknown	287.2	269.1		+	−
31	44.4	Unknown	307.2	289.0, 235.0, 185.0		+	−
32	51.0	Unknown	309.2	291.1, 208.9, 184.9		+	−
33	53.9	Unknown	487.5	469.4		+	−

**Table 3 molecules-27-01613-t003:** Antioxidant activities of hydroalcoholic extracts of *L. angustifolia* and *L. × intermedia* expressed as the mean ± standard deviation milligrams of positive control per gram of lavender biomass. The results were obtained from three independent experiments on the replicates of the extracts.

	LA	LI
Antiradical activity	94.17 ± 6.29 mg eqT/g	94.51 ± 2.85 mg eqT/g
Chelation activity	1.49 ± 0.03 mg eqEDTA/g	2.10 ± 0.13 mg eqEDTA/g
Fe3+ reduction capacity	89.36 ± 3.92 mg eqAA/g	74.53 ± 2.74 mg eqAA/g

T, Trolox; AA, ascorbic acid.

**Table 4 molecules-27-01613-t004:** Acetylcholinesterase (AChE) and tyrosinase inhibition activities of *L. angustifolia* (LA) and *L. × intermedia* extracts, and reference inhibitors galantamine and kojic acid. The results are expressed as the mean ± standard deviation of IC50 values. The results were obtained from three independent experiments on the replicates of the extracts.

	AChE	Tyrosinase
LA	5.35 ± 0.47 mg/mL	5.26 ± 0.02 mg/mL
LI	6.67 ± 0.12 mg/mL	6.56 ± 0.16 mg/mL
Galantamine	18.83 ± 1.05 μg/mL	-
Kojic acid	-	18.13 ± 0.45 μg/mL

## Data Availability

The data presented in this study are available on request from the corresponding author.

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
