# Peer review of "Characterization and Valorization of the Agricultural Waste Obtained from *Lavandula* Steam Distillation for Its Reuse in the Food and Pharmaceutical Fields"

_molecules, 2022, doi:10.3390/molecules27051613_

Round 1
Reviewer 1 Report
Comments and Suggestions for Authors
The aim of this paper is to describe the Characterization and valorization of the agricultural waste obtained from Lavandula steam distillation for its reuse in the food and pharmaceutical fields.
The idea is not so much novel, however the obtained data are presented in scientific way and the paper is well written and scholarly informative. Hence the reviewer thinks that this paper deserves to be published, but after some corrections:
- If possible provide the authors should provide the FITR spectrum.
2. The authors should provide the screening test data for phenolic and flavonoids.
Minor corrections
- The authors should correct the first line of abstract such as " The main focus of the current/present research"
- in Introduction line 4 more than 700 million " Mg" should be "mg".
- 30-32 % reference from last five years, i think > 40 % is better from last 5 years.
Author Response
- If possible provide the authors should provide the FITR spectrum
The authors believe that the addition of the FTIR spectra of the extract might not be useful for the research scope, since the project aimed at the evaluation of the main composition and biological activities of the extracts.
- The authors should provide the screening test data for phenolic and flavonoids.
The results of the calibration curves with gallic acid and quercetin for the determination of TPC and TFC were provided in the supplementary materials.
Minor corrections
1.The authors should correct the first line of abstract such as " The main focus of the current/present research"
We corrected the sentence.
2. In Introduction line 4 more than 700 million " Mg" should be "mg".
“Mg” is not a mistake, “M” (Mega) was used as multiple of grams. However, to avoid misunderstandings, “Mg” was replaced by “tons”.
3. 30-32 % reference from last five years, i think > 40 % is better from last 5 years.
The references were updated.
Reviewer 2 Report
Comments about the manuscript molecules-2022- 1591485
Characterization and valorization of the agricultural waste obtained from Lavandula steam distillation for its reuse in the food and pharmaceutical fields
This manuscript described the chemical characterization (oil-exhausted biomass and hydrosol) and some bioactivities (oil-exhausted biomass) of the by-products from the steam distillation of the species Lavandula angustifolia and L. x intermedia aerial parts. It involves two species with various properties and industrial and health applications, addresses an interesting topic, but some important considerations must be discussed.
The main consideration to be made is how the chemical part of the work (especially EOs and phenolic compounds) differs from those found in other publications involving both species and with similar objectives, approaches, and results. Why did the authors choose this subject despite there being several publications with both species with similar objectives? I think some discussions are incomplete. Some references are missing.
Authors should review the citations (including format) and their corresponding references at least from line 123 onwards. In some cases, there is no correlation between citations and reference numbers. In addition, they should consider carefully revising the format of some references. Some names of botanical species are not written in italics and other details.
Results and Discussion
2.1. Total phenolic and total flavonoid content
Lines 105-107: “Preliminary studies selected the mixture of ethanol and water (50:50) as the optimal solvent for the extraction of polyphenols from the oil-exhausted biomasses.”
Are these aforementioned studies published?
Conclusion
Lines 433-435: “The developed method of extraction of the oil-exhausted biomasses proved to be a promising strategy for the recovery of polyphenols by using food-grade solvents (ethanol and water).”
I didn't find in the Method section the development of the mentioned extraction method.
Material and Methods
3.4. Total polyphenolic and flavonoid content
Lines 312-313: “The total polyphenolic content (TPC) was determined by Folin–Ciocalteu method on the three different extracts obtained from each Lavandula oil-exhausted biomass.”
Lines 321-323: “The total flavonoid content (TFC) was determined according to the aluminum chloride method on the three different extracts obtained from each Lavandula oil-exhausted biomass.”
It was unclear whether the authors performed a triplicate of each extraction (nine samples x 2) or one analysis of each extraction (three samples x 2). Please explain.
Some minor revisions should be made, such as:
Table 2
Please consider reviewing: “Luteolin 7-O-diglucoronide” to Luteolin 7-O-diglucoronide (italic), the same revision must be extended to the nomenclature of the other flavonoid glycosides in the Table.
Lines 167-189 and 345: Please consider reviewing: “m/z” to m/z (italic)

Author Response
Characterization and valorization of the agricultural waste obtained from Lavandula steam distillation for its reuse in the food and pharmaceutical fields
This manuscript described the chemical characterization (oil-exhausted biomass and hydrosol) and some bioactivities (oil-exhausted biomass) of the by-products from the steam distillation of the species Lavandula angustifolia and L. x intermedia aerial parts. It involves two species with various properties and industrial and health applications, addresses an interesting topic, but some important considerations must be discussed.
The main consideration to be made is how the chemical part of the work (especially EOs and phenolic compounds) differs from those found in other publications involving both species and with similar objectives, approaches, and results. Why did the authors choose this subject despite there being several publications with both species with similar objectives? I think some discussions are incomplete. Some references are missing.
The present work aimed to characterize the waste obtained from the extraction of lavender EOs. The properties of the extracts of lavender aerial parts are well-known. However, the majority of the studies present in the literature did not consider the oil-exhausted biomasses. Thus, to promote the reuse and the valorization of this agricultural waste is necessary to focus the study on the oil-exhausted biomasses. Indeed, during the steam distillation process, the plant is subjected to high temperatures for a long time, resulting in the partial degradation of bioactive compounds, being polyphenols thermolabile compounds. The present work, to the best of our knowledge, was the first study of the enzyme inhibitory effects of extracts obtained from lavender biomass after the steam distillation process.
We agree that this aspect of the work was not sufficiently emphasized. For this reason, the beginning of the result and discussion section and the conclusions were increased.
Authors should review the citations (including format) and their corresponding references at least from line 123 onwards. In some cases, there is no correlation between citations and reference numbers. In addition, they should consider carefully revising the format of some references. Some names of botanical species are not written in italics and other details.
- We apologize for the mistakes in numbering the references. We updated the number of the references and the name of the botanical species in italics.
Results and Discussion
2.1. Total phenolic and total flavonoid content
Lines 105-107: “Preliminary studies selected the mixture of ethanol and water (50:50) as the optimal solvent for the extraction of polyphenols from the oil-exhausted biomasses.”
Are these aforementioned studies published?
No, these are previous studies for solvent screening that were not published. We specified this aspect in both the method and result sections.
Section 3.3: The extraction was performed by dynamic maceration, and several extracting solvents or solvent mixtures were tested (ethanol, methanol, water, ethyl acetate, and hydroalcoholic solutions at different ratios). The optimal solvent was selected based on the efficiency in recovering polyphenols, quantified by Folin–Ciocalteu method as described below. Briefly, 3.5 grams of oil-exhausted aerial parts were extracted by dynamic maceration with 40 mL of the extracting solvent. The solution was filtered into a volumetric flask and the biomass was extracted two times more with 35 mL of the same solvent. The filtrates were adjusted at the final volume of 100 mL and stored at 4 °C. The extraction was performed in triplicate. For the enzymatic assays, the ethanol was removed under vacuum and the remaining aqueous suspension was freeze-dried (Lio 5P, CinquePascal, Milan, Italy). The extracts obtained by the optimal solvent for the recovery of polyphenols were used for the following analyses.
Section 2.1: The solvent screening for the recovery of polyphenols from oil-exhausted lavender biomasses suggested that the mixture of ethanol and water (50:50) was the optimal solvent in terms of TPC. The yields of the dry extracts were determined by freeze-drying the liquid extracts, obtaining 211.5 ± 7.8 mg/g e 193.0 ± 2.6 mg/g of dry extract of biomass for LA and LI respectively.
Conclusion
Lines 433-435: “The developed method of extraction of the oil-exhausted biomasses proved to be a promising strategy for the recovery of polyphenols by using food-grade solvents (ethanol and water).”
I didn't find in the Method section the development of the mentioned extraction method.
We agree with the reviewer’s comment since we don’t show any result regarding the preliminary studies of extraction. For this reason, we changed the sentence as follows:
“The selected method of extraction of the oil-exhausted biomasses proved to be a promising strategy for the recovery of polyphenols by using food-grade solvents (ethanol and water), also applicable on large scale.”
Material and Methods
3.4. Total polyphenolic and flavonoid content
Lines 312-313: “The total polyphenolic content (TPC) was determined by Folin–Ciocalteu method on the three different extracts obtained from each Lavandula oil-exhausted biomass.”
Lines 321-323: “The total flavonoid content (TFC) was determined according to the aluminum chloride method on the three different extracts obtained from each Lavandula oil-exhausted biomass.”
It was unclear whether the authors performed a triplicate of each extraction (nine samples x 2) or one analysis of each extraction (three samples x 2). Please explain.
The following sentence was added: “The results were expressed as mean ± standard deviation calculated from the results obtained in duplicate for each replicate of extract (n=3).”
Some minor revisions should be made, such as:
Table 2
Please consider reviewing: “Luteolin 7-O-diglucoronide” to Luteolin 7-O-diglucoronide (italic), the same revision must be extended to the nomenclature of the other flavonoid glycosides in the Table.
Lines 167-189 and 345: Please consider reviewing: “m/z” to m/z (italic)
All the suggestions were taken into account and italic was used.
Reviewer 3 Report
The submitted work raises an interesting topic concerning the use of biomass produced from Lavandula steam distillation. The authors show that this product can be an additional source of phenolic and flavonoid compounds that can still be used, for example, in the food, cosmetic and pharmaceutical industries. The content of the analyzed substances is so high that it is possible to look for new solutions, especially in the aspect of environmental protection. The authors discussed the literature data on the discussed subject, showing how many factors (environmental, laboratory) affect the quality of the obtained biomass. They proposed a relatively simple method of extracting the analyzed compounds with the use of safe solvents for the environment. Overall, the work is interesting and valuable. However, I would supplement it with some information on the validation of the LC-MS/MS and MS2 method used to determine the content of active compounds (including method scope, LOD, LOQ, precision, accuracy).
Author Response
Dear reviewer, the validation of the LC-MS/MS and MS2 method is missing because we did not perform the quantification of the active compounds. The LC-MS/MS was employed for qualitative purposes.